# Intratumoral Heterogeneity and Metabolic Cross-Feeding in a Three-Dimensional Breast Cancer Culture: An In Silico Perspective

**DOI:** 10.3390/ijms252010894

**Published:** 2024-10-10

**Authors:** Jorge E. Arellano-Villavicencio, Aarón Vázquez-Jiménez, Juan José Oropeza-Valdez, Cristian Padron-Manrique, Heriberto Prado-García, Armando R. Tovar, Osbaldo Resendis-Antonio

**Affiliations:** 1Human Systems Biology Laboratory, Instituto Nacional de Medicina Genómica (INMEGEN), Mexico City 14610, Mexico; jorge.arellano.bioexp@gmail.com (J.E.A.-V.); vazqaaron@gmail.com (A.V.-J.); cristianjuliocesar.agualimpia@gmail.com (C.P.-M.); 2Programa de Doctorado en Ciencias Bioquímicas, Universidad Nacional Autónoma de México (UNAM), Mexico City 04510, Mexico; 3Centro de Ciencias de la Complejidad, Universidad Nacional Autónoma de México (UNAM), Mexico City 04510, Mexico; juan.oropeza@c3.unam.mx; 4Programa de Doctorado en Ciencias Biomédicas, Universidad Nacional Autónoma de México (UNAM), Mexico City 04510, Mexico; 5Laboratorio de Onco-Inmunobiologia, Departamento de Enfermedades Crónico-Degenerativas, Instituto Nacional de Enfermedades Respiratorias Ismael Cosio Villegas (INER), Mexico City 14080, Mexico; hpradog@yahoo.com; 6Departamento de Fisiología de la Nutrición, Instituto Nacional de Ciencias Médicas y Nutrición Salvador Zubirán (INCMNSZ), Mexico City 14080, Mexico; armando.tovarp@incmnsz.mx; 7Coordinación de la Investigación Científica—Red de Apoyo a la Investigación, Universidad Nacional Autónoma de México (UNAM), Mexico City 14080, Mexico

**Keywords:** cancer metabolism, tumor microenvironment, heterogeneity, MCF-7, reverse Warburg effect, lactate, systems biology

## Abstract

Today, the intratumoral composition is a relevant factor associated with the progression and aggression of cancer. Although it suggests a metabolic interdependence among the subpopulations inside the tumor, a detailed map of how this interdependence contributes to the malignant phenotype is still lacking. To address this issue, we developed a systems biology approach integrating single-cell RNASeq and genome-scale metabolic reconstruction to map the metabolic cross-feeding among the subpopulations previously identified in the spheroids of MCF7 breast cancer. By calibrating our model with expression profiles and the experimental growth rate, we concluded that the reverse Warburg effect emerges as a mechanism to optimize community growth. Furthermore, through an in silico analysis, we identified lactate, alpha-ketoglutarate, and some amino acids as key metabolites whose disponibility alters the growth rate of the spheroid. Altogether, this work provides a strategy for assessing how space and intratumoral heterogeneity influence the metabolic robustness of cancer, issues suggesting that computational strategies should move toward the design of optimized treatments.

## 1. Introduction

The Warburg effect, described by Otto Warburg in 1956, has been one of the most paradigmatic metabolic hallmarks in the study of cancer, where tumor cells oxidize high rates of glucose, as an energy source, to bio-transform it into lactate, even in the presence of oxygen [1]. Consequently, this transformation acidifies the tumor microenvironment and favors cell transformation and invasion into other tissues [2]. However, some tumors adopt different strategies to obtain energy. For instance, breast cancer cells incorporate glutamine [3] to adapt to alterations in their microenvironment and compensate for their energy demand. Furthermore, there is evidence suggesting that this type of cancer oxidizes fatty acids, whose metabolic activity is closely related to survival and resistance to chemotherapeutic treatments [4]. In recent years, the metabolic communication between tumors and adjacent cells, by shaping the local microenvironment, has gained interest due to its role in metabolic robustness and tumor survival. Experimentally, this metabolic interdependence has been observed in cancer-associated fibroblasts (CAFs) and cancer cells [5,6]. This relationship has been hypothesized to be supported by the production of reactive oxygen species (ROS) by cancer cells and its delivery into the medium. In turn, these conditions result in elevated glycolytic activity in CAFs, the production of lactate, and its excretion to the extracellular medium. This situation benefits tumor cells, which can dispose of the lactate released into the extracellular medium and respond to unfavorable conditions such as hypoxia or changes in pH. This association has been described as a reverse Warburg effect, a feature accepted as a critical metabolic adaptation in the progression of pathology [5,6]. The transient nature of this association and the lack of information on the conditions and metabolic regulatory pathways involved in its establishment add significant complexity to its study.

Today, one factor that evidences the complexity of cancer is the intertumoral (between tumors) and intratumoral (within the tumor) metabolic heterogeneity [7]. The progress of omics technologies, such as single-cell and spatial transcriptomics, has opened a new window to understanding tumor phenotypes and heterogeneity using gene expression data [8]. For instance, the implementation of single-cell RNAseq technology (scRNA-Seq) into an in vitro 3D culture of luminal invasive breast cancer subtype A (MCF-7) determined the coexistence of three cell subpopulations in the spheroid, each one with proliferative, invasive, and reservoir phenotypes [9]. Some questions emerge: What type of metabolic interdependence exists between the heterogeneous composition of cancer? What metabolites can be shared between the subpopulations? How can we modulate the medium to interfere with its progression and survival?

Gene Set Enrichment analyses of sequencing data have significantly contributed to the characterization of specific functions in tumor cells and non-tumor tissues, providing relevant information for prognosis and the development of targeted therapies [10]. However, this information does not determine whether functional heterogeneity is supported by metabolism or how cells may couple to promote tumor survival. The cooperative metabolic landscape that may be present inside a solid tumor and the complexity attributed to its cellular heterogeneity open the door to implementing comprehensive strategies where mathematical formalism and bioinformatics tools could provide approximations with the capacity to be experimentally corroborated.

However, computational tools for generating genome-scale metabolic reconstructions (GEMs) from expression data have been previously suggested in bulk and single-cell transcriptomics to obtain better-calibrated tissue-specific reconstructions with the ability to correlate with experimentally measured metabolites [11,12,13,14]. Systems biology and metabolic engineering have been useful in simulating the specific production of a metabolite or stimulating cell growth by implementing computational formalisms in a steady-state condition [15]. In particular, the metabolic modeling of GEMs through a flux balance analysis (FBA) is a valuable formalism to predict the growth rates of some organisms and explore in silico experimental strategies to maximize the production of a metabolite of interest [16].

In the beginning, GEM-based models for cancer studies were limited to representing a single specific biological trait and modeling tumor cells at an average population level [17]. At present, constraint-based modeling can explore the metabolic phenotype not only with bulk data of cancer cells but also with different GEMs comprising communities of cells or subpopulations. This advancement stems from data obtained from sc-RNAseq datasets [18], which have highlighted potential connections between various cell groups within a solid tumor. These studies have confirmed the importance of lactate in phenomena such as the reverse Warburg effect, where its knockout affects tumor growth. However, in heterogeneous environments, we often overlook factors that could introduce noise into the results, which are often challenging to analyze. In the case of solid tumors, cell populations may exist in varying proportions within the tumor, creating small local microenvironments where they engage in their exchanges or even compete metabolically. Some tools, such as MICOM (MIcrobial COMmunity), have been implemented and validated in modeling complex communities, allowing for a more precise analysis of interactions between individuals and extracellular microenvironments [19]. This is achieved by using a framework that enables the prediction of dynamic flux distributions based on community composition and environmental conditions, particularly to evaluate how a cell might respond to the variable conditions of its environment. However, the effectiveness of these models heavily depends on the quality of the transcriptome data entered, as it opens the door to false negatives in the results. To reduce this effect, tools such as sc-PHENIX [20] have improved transcriptome data preprocessing, reducing excessive data smoothing and enabling a better characterization of cellular phenotypes.

In this paper, we propose an alternative pipeline for the in silico metabolic modeling of a heterogeneous population of cancer cells characterized by scRNAseq data. We show that this formalism has some advantages over previous contributions. Among them, our formalism can include the relative concentrations of the subpopulation inside the tumor, an important factor in obtaining the biomass function of the entire community. To verify our framework, we applied it to describe the metabolic activity of spheroids of MCF-7 at two time points of progression, days 6 and 19. As reported previously, the spheroids were composed of three subpopulations (invasive, proliferate, and reservoir), and the media were perfectly controlled in the lab. By selecting scRNAseq data for each subpopulation at both times, we accomplished a metabolic reconstruction and simulated the metabolic activity at a community level. Community modeling confirmed the cooperation between subpopulations and highlighted the presence of phenomena such as the Warburg effect and reverse Warburg effect in invasive phenotypic subpopulations that persist throughout spheroid growth. Additionally, it demonstrated the metabolic compensation on which certain subpopulations rely to continue growing by incorporating alternative carbon sources and phenomena such as the secretion of oncometabolites, which could serve as signaling molecules orchestrating cellular transformation. The consequence of a gradient of oxygen, the map of possible metabolic crosstalk among the subpopulation, and the prediction of metabolic targets for controlling the growth rate of the tumor are surveyed at the end of this paper.

## 2. Results

### 2.1. Community Metabolic Modeling Strategy Design

To model the metabolism of a cell community, we developed a strategy that involved creating GEMs based on scRNA-seq data obtained from a spheroid culture of MCF-7. This approach enabled us to generate three reconstructions representing distinct cell subpopulations: invasive, reservoir, and proliferating subpopulations (Figure 1a). Next, to address the challenge of constructing a community model from these metabolic reconstructions, we proposed the implementation of tools capable of integrating the reconstructions by specifying a growth medium and the relative proportions of each subpopulation. This setup was designed to simulate two temporal stages of cell spheroid growth (days 6 and 19) to elucidate the impact of cell distribution, subpopulation proportions on community growth, and the exchange fluxes with the microenvironment, as illustrated in Figure 1b.

### 2.2. Specific Subpopulation Genome-Scale Metabolic Reconstruction from scRNAseq

To survey the metabolic cross-feeding among the cancer subpopulation in MCF7, we obtained a genome-scale metabolic reconstruction starting from the scRNA-Seq data associated with the three identifiable subpopulations at two times of the growth process: on days 6 and 19 of culture. As originally discovered by the authors, these subpopulations have different biological functions (proliferative, invasive, and reservoir functions), and the interactive crosstalk among them is presumably an essential factor stabilizing the spheroid. We preserved the original annotation made by the authors. Before proceeding with the metabolic reconstructions, an imputation method for the scRNA-seq data was applied to the count matrix to denoise the data and correct the ‘dropout events’. This minimizes the noise characteristic of scRNA-seq technology [20].

Overall, the imputation contributed to improving the subsequent metabolic reconstruction procedure. After imputing the count matrix, we proceeded with the reconstruction by applying the Cost Optimization Reaction Dependency Assessment (CORDA) method to the gene expression profiles of each subpopulation. Despite the variety of techniques used to accomplish such a reconstruction, we selected the CORDA due to its ability to create non-minimalistic metabolic models with a greater fidelity in representing biological characteristics [9,13,21]. As a result, we created three metabolic reconstructions, one for each cell subpopulation described inside the spheroid. As expected, each reconstruction had distinctive metabolic capabilities reflecting the feasible space of functionalities in each subpopulation; see Appendix A. To ensure the quality of these reconstructions and their nearness to represent actual metabolic processes, we evaluated each network through the Metabolic Modeling Test (MEMOTE) (see Section 4), a computational platform used to assess the quality of genome-scale metabolic reconstructions through a variety of parameters such as the consistency of the stoichiometric matrix, biomass production, and the gene–protein–reaction association. The quality report obtained with the MEMOTE indicated that the reconstructions had a subtotal consistency score of 95%, suggesting that they presented a proper stoichiometry matrix, mass balance, charge balance, and adequate connectivity between metabolites; see Appendix A.

### 2.3. Optimization and Community Modeling

Once we obtained and proved the quality of each reconstruction, we proceeded to computationally simulate the metabolic capacities when coexisting as a system. We tested the community at each time point, on days 6 and 19 of MCF-7 culture (the days that were analyzed in the experimental design of Muciño et al. [9] and when constrained by the medium (see Figure 1b). This model allowed us to tackle two fundamental challenges: postulating the shared metabolites between the subpopulations (cross-feeding metabolic activity) and evaluating how their communication affects the metabolic stability and biomass production of the entire community. All reconstructions were integrated and modeled in MICOM [19].

MICOM considers the relative abundance of each subpopulation and the metabolites in the extracellular medium to maximize a community biomass function. Computationally, MICOM conducts a flux balance analysis by integrating linear optimization with an L2 regularization constraint to estimate the biomass required to maintain the biomass for each population (see Section 4—Community Modeling—MICOM). To reproduce the experimental conditions, we defined the extracellular medium in terms of the metabolic composition of the culture media used to grow the MCF7 spheroids in the original experiment experimentally; see Appendix A. To simplify our analysis and avoid diffusion processes, we considered that all the media resources were available for each subpopulation at both simulation times (6 and 19 days of progression of the spheroid). Under these assumptions, our metabolic simulation indicated that the community growth rate was 0.026 mmol/cell/h, corresponding to a doubling time of approximately 26 h on both days; for details on the duplication time calculations, see Appendix A. Consistently with our in silico estimations, this doubling time is nearly associated with the experimental measurements reported for the MCF-7 cell line between a 20 and 24 h doubling time [22].

At this point, our model was used as a computational platform to create testable hypotheses around the exchanged metabolites with the medium and the metabolic cross-feeding between the subpopulations. This interaction can be represented graphically as groups corresponding to the subpopulations with the respective exchange interactions between them and with the extracellular medium; see Figure 2a. This computational platform can be used to guide design experiments, prove or generate biological hypotheses, and explore strategies for searching for metabolic targets for potential therapeutic interventions.

### 2.4. Effect of Pseudo-Spatial Distribution of Subpopulations in MCF-7 Spheroid

In the previous section, we simulated the metabolic interaction among the three subpopulations by neglecting the diffusion process inside the tumor spheroids. However, as revealed by spatial RNAseq, the cellular distribution and arrangement of metabolites in different tumor strata (such as oxygen and carbon sources) significantly influence the intratumoral profiles of metabolic heterogeneity [23]. To emulate this process and quantitatively assess the effect of metabolic gradients on tumor malignancy and stability, we extend our computational genome-scale metabolic modeling by including layers of gradients over oxygen, a metabolite whose concentrations inside the tumor depend exclusively on the size of the spheroid. In the rest of this work, we assume that oxygen is a metabolite consumed solely by the environment and a key metabolite in establishing intratumoral cellular heterogeneity.

As a simplification, we considered only a radial, one-dimensional distribution of the oxygen gradient defined by three discrete levels in the inner microenvironment. The simulation started from the outside to the center of the spheroid: the first level emulated cell normoxia, the second level represented an intermediary level of oxygen abundance, and the last level indicated a hypoxic condition. Previous results have shown that each subpopulation triggers unique transcriptional and possibly metabolic pathways [9]. For instance, the invasive subgroup triggers a response to hypoxia, KRAS, and matrix remodeling pathways, typically associated with tumor cells deprived of oxygen. Similarly, proliferative cells activate metabolic pathways related to cell division and biomass production. These facts postulate that each subpopulation might have different levels of oxygen accessibility. Therefore, as an initial supposition, we modeled the spheroid’s geometry by layers. Given the uncertainty around the precise location of the cell subpopulations in the spheroid, we modeled the metabolic interaction over six different spatial configurations, each defined by different spatial arrays of subpopulation and oxygen availability (see Figure 2b). In principle, all these configurations are feasible, at least in an in silico model; however, we focused on the configurations that maximize the growth rate of all the communities to survey the possible metabolic communication between the subpopulations. Hence, we calculated the community growth rate for each spatial distribution, considering the abundance of the subpopulations previously reported on day 6 or 19. Figure 2c,d show the community’s growth rate; the results show the cellular organization that the spheroid could have on both days of study when taking into account the best growth rate of the community.

Consequently, the spatial configuration denoted by condition E in Figure 2c was the ideal combination to optimize the doubling time of the spheroid on day 6. According to intuition, this configuration suggests that the invasive subpopulation is located in the center of the spheroid, the reservoir is located at a medium level, and the proliferative subpopulation is located on the outer layers. This optimized distribution suggests that the invasive cells are better adapted in the center of the spheroids, a place with a relatively low uptake of oxygen. In contrast, when changing the corresponding population abundances for day 19 of growth, spatial configuration E (see Figure 2d) proved the least optimal. Our simulations suggest that placing the invasive cells on the outside enhances community growth (see Figure 2d condition A). According to the single-cell characterization of the spheroids [9], the diameter increases linearly, suggesting that the proliferation rate should be constant over time, which might be related to the E6 and A19 conformations. Thus, we conclude that these configurations provide metabolic advantages to optimize the community growth rate at a systemic level.

To analyze the metabolic phenotypes that emerged from these configurations, the following sections (1) evaluate the metabolic exchange between the environment and the tumor, as well as the metabolic cross-feeding among populations, and (2) identify the metabolic pathways involved in each subpopulation, based on enzymatic reactions classified through the Virtual Metabolic Human (VMH) database [24].

### 2.5. Metabolic Mapping of Cancer Spheroids: Secretion, Consumption, and Cooperation among Subpopulations

Having simulated the growth rate of the spheroid at both time points of progression, we explored how the metabolic exchange between the tumor and the extracellular milieu is influenced by the cellular organization of the spheroid at both these times (see Section 4).

To simplify data presentation, we generated a Boolean representation of metabolic excretion or consumption between the subpopulations and the environment. The metabolites could be classified into three categories: those secreted and produced simultaneously and those that alternate between production and excretion as time goes on. For example, we noted that most of the amino acids and carbon sources, such as galactose and glutamine, were consumed by each subpopulation from the media at both times (see Figure 3a). Meanwhile, our modeling suggests that metabolites such as carbon dioxide, ammonia, and succinate were excreted from the community to the media. Furthermore, we identified a third class of metabolites whose production consumption changed as time varied. This last category included 2-oxoglutarate (AKG), acetoacetate, fumarate, L-lactate, and amino acids such as glycine and L-alanine.

In terms of the metabolic cross-feeding among subpopulations, the results showed that all three subpopulations had metabolites present in the medium; however, some metabolites were not initially found in the culture medium, such as those that are metabolic byproducts of one of the subpopulations. Figure 3a shows all the metabolites produced by one subpopulation and consumed by another. Among these metabolites, we primarily identified essential amino acids like L-isoleucine, L-leucine, L-lysine, L-methionine, L-phenylalanine, L-threonine, L-tryptophan, and L-valine; non-essential amino acids like L-asparagine, L-histidine, and L-serine; conditional non-essential amino acids like L-arginine, L-cysteine, L-glutamine, and L-tyrosine; and carbohydrates such as galactose. Our model postulated that these metabolites maintained constant consumption regardless of the growth day or subpopulation type. However, some amino acids did not follow this behavior. For example, glycine, a metabolite consumed by the invasive population, could be taken from the medium and reservoir cells (see Figure 3a). However, proliferative subpopulations could also contribute to this amino acid on day 6. Furthermore, we observed that specific metabolites were produced exclusively by one type of subpopulation. For instance, the addition of alanine to the medium was regulated by the invasive cells on both days of study, similar to proline by the proliferative cells.

As Figure 3b shows, our analysis postulates an intricate metabolic mapping underlying an interdependence between the cancer subpopulations in MCF7 spheroids. The production and consumption of metabolites crosslink to meet the metabolic requirements of the growing community. Figure 3b shows the subpopulation from which the metabolites originate and where they are required. Such is the case of alpha-ketoglutarate, fumarate, and lactate, whose consumption and secretion profile change at both examined times of the spheroid. The panels in Figure 3b,c show the cross-feeding metabolic profile of the community on days 6 and 19. On day six, the proliferative cells secrete alpha-ketoglutarate and lactate, a characteristic marker of the Warburg effect. In contrast, the invasive cells dispose of this metabolite by incorporating it into their pathways. This effect is also observed on day 19 and can be seen as a hallmark between the proliferative and invasive subpopulations. Altogether, the cooperative metabolic profile between the three subpopulations creates a complex scenario of metabolic inner/outer communication at different stages of growth. Among these possibilities, some metabolites behave depending on the day of growth. For instance, AKG is secreted to the medium on day six by the proliferating cells. On day 19, the invasive cells show this behavior, exchanging roles between excretion and consumption only between these two subpopulations. By analyzing a portion of the top 20 of the approximately 350 exchange reactions and 2000 community reactions, we may be ruling out the possibility that AKG-like switches are also present with metabolites whose exchange fluxes are minimal. These scenarios could add a touch of complexity to the design of metabolic targets due to the metabolic adaptation and reconfiguration of a spheroid over time.

To assess metabolic adaptations, we estimated the metabolic pathways with the highest activity in each subpopulation. Then, based on the reconstruction of the metabolic pathways, reactions with non-zero flux units (in any directionality) were isolated to determine each pathway’s importance in each subpopulation. A common feature is that all three subpopulations had activity in their main pathways of central metabolism, such as glycolysis, the TCA, and the pentose phosphate pathway (see Appendix A). However, there were cases, such as lipid metabolism (fatty acid oxidation and fatty acid synthesis), where a higher number of positive-flux reactions was observed in the proliferative and reservoir cells than in the invasive cells. On day 19, the invasive cells had more active reactions due to changes in spheroid abundance and size (see Appendix A).

Despite this analysis supplying a proxy of the metabolic activity associated with each subpopulation, the reactions present in each pathway usually cannot be the same; however, if so, the behavior of the directionality of the metabolic fluxes may vary in those reactions with reversible capacities. The description of the directionality changes in the reactions provided a more comprehensive approach to tracking how each subpopulation uptakes, transforms, and utilizes the metabolites in the medium for energy generation and biomass production.

### 2.6. Metabolic Activity Inside the Populations

Once the central metabolic pathways were identified, they were used to analyze individual behavior and determine the existence of metabolic cross-feeding between subpopulations. Among these, we identified energy pathways such as glycolysis, the Krebs cycle, oxidative phosphorylation, the pentose phosphate pathway, and some other metabolic pathways that might be of interest (see Figure 4 and Appendix A). The simulation was not only limited to providing fluxes of exchange within the medium but also contained fluxes of intracellular reactions of each subpopulation. This information allowed us to generate a descriptive picture of the behavior of each cell type—for example, the glycolytic activity of the community. By having galactose as the main carbohydrate according to the original experimental design, we consequently observed high activity of the enzyme galactosidase in most of the subpopulations on both days of study, except for in the invasive cells, in which this enzyme showed lower fluxes of activity on day 6 of growth (see Figure 4b)—suggesting that this metabolite is not as essential for this subpopulation in early periods of growth.

The information obtained by simulation suggests that the invasive subpopulation does have glycolytic activity, as shown in Figure 4a, as metabolic fluxes are seen in most of the enzymes involved in glycolysis. However, particular details such as lactate dehydrogenase (LDH) enzyme activity do not reflect a directionality similar to the proliferative cells that initially convert pyruvate to lactate but instead that the invasive cells are in a reversal process of this action and not by carrying out the oxidation of galactose to pyruvate by a classical pathway. Reaction directionality reversal behaviors further solidify the association theory between subpopulations by incorporating metabolites secreted into the medium by neighboring cells.

The origin of metabolites by alternative pathways, such as anaplerotic pathways, often converges to the TCA for generating reducing power and oxidative phosphorylation. Given the results shown in Figure 4c, we concluded that full participation of the TCA is not necessary to satisfy the energy demand of the cells. Although all reconstructions have full TCA, the optimization suggests a preference for certain enzymes that compensate for the objective function of the simulation. For example, given the directionality of the reaction, the model suggests that succinate coenzyme A ligase, one of the critical enzymes of the TCA, consumes succinate to produce succinate and GTP from succinyl-CoA. These reactions could be preceded by the activity shown by alpha-ketoglutarate dehydrogenase, which provides the succinyl-CoA originating from the presence of alpha-ketoglutarate (see Figure 4c). Both enzymes are shared among the three subpopulations. However, particular cases, such as the invasive cells on day 19 and proliferating cells on day 6, activate another section of the TCA, such as succinate dehydrogenase, whose reversibility converts fumarate, which might be taken up from the extracellular medium when secreted by reservoir cells to succinate (see Figure 4c).

The partial recovery of pathways, such as the TCA, suggests that the availability of metabolites such as pyruvate for energy may have different origins, ranging from galactose oxidation or the incorporation of lactate from the microenvironment to the involvement of anaplerotic pathways such as alanine incorporation, of which L-alanine transaminase activity was recorded in the invasive and reservoir subpopulations (see Figure 4d). The simulation gave us a broader picture of what alternative energy pathways the cells might use. The simulation also facilitates the analysis of alternative anaplerotic pathways. To identify alternative pathways, we separated the already described features of tumor cell metabolism, such as the incorporation and activity of enzymes like mitochondrial glutaminase and glutamate dehydrogenase enzymes, which coordinate to generate alpha-ketoglutarate in the TCA; see Figure 4d.

This set of results allowed us to characterize each subpopulation and assign specific metabolic characteristics. According to Muciño et al., 2024 [9], the cells have glycolytic activity. Our results complement this assertion by providing the variations in each subpopulation to obtain energy. These differences could be explained by the capacity of each cell to adapt its metabolism to the conditions of its microenvironment and the cooperation between individuals in the community. Under this criteria, we explored the adaptive capacities of the community in response to stress in its microenvironment. Oxygen was directly taken as an essential element in the community that could quickly stimulate a change in the subpopulations. As we argue in the next section, the action of different oxygen zones makes evident the importance of some amino acids for the survival of the whole community.

### 2.7. Metabolic Response of Cancerous Subpopulations under Gradients of Oxygen

The stress induced in the community through the oxygen supply of each subpopulation could be an essential factor for the appearance of heterogeneous profiles in the spheroid; this is because one of the variables that allowed us to establish cooperation among the subpopulations was the presence of a simulated gradient in the community model. To evaluate the community’s response to adverse scenarios, such as hypoxia, a scenario was simulated where the environmental conditions changed due to oxygen availability. This was also carried out to determine the ability of each subpopulation to adapt to change and to elucidate cellular cooperation to ensure the survival of all cells (see Figure 5). The response was reflected in a greater or lesser preference for selected metabolites as possible energy sources, such as galactose, pyruvate, L-lactate, and AKG, as well as amino acids such as glycine, L-alanine, L-glutamine, and L-serine.

Taken as an example is the consumption of glutamine, an amino acid that is associated with the fundamentals of tumor metabolism and has an essential role as an alternative carbon source for tumor cells to ensure survival under unfavorable conditions. This metabolite proved crucial for the three subpopulations on both study days. The simulation showed that the proliferating subpopulation had the most glutamine in the medium on day six. As oxygen availability decreased, the invading cells began to increase their consumption. The opposite was the case on day 19, where the invading cells had most of the glutamine, and this behavior did not appear to be affected by the oxygen gradient (see Figure 5a,b).

The cells could incorporate metabolites in the medium, such as pyruvate, to obtain energy under normoxic conditions or in response to gradients. The proliferating cells disposed of pyruvate on day 6 (see Appendix A). As oxygen decreased, this behavior was maintained, as well as in the reservoirs but in a lower proportion. By day 19, under normoxic conditions, the entire community had a low amount of this metabolite, which could be interpreted as a low preference or essentiality. However, as disposition decreased, the invasive subpopulation began to mobilize pyruvate into its interior (see Appendix A). The effects differed significantly from those observed when consuming the primary carbon source, galactose. An almost linear gradient accompanied the decrease in oxygen on both days (see Appendix A) as a response to oxygen availability and possibly by incorporating alternative sources. Another type of response observed in the simulation was cooperation between the subpopulations; under normoxic conditions, there was already an association between the proliferating and invasive cells and lactate mobility. However, on day six, the decrease in oxygen increased the transport of lactate into the invasive cells from the proliferating cells and the reservoir. Stages close to zero activated lactate production in the reservoir and contributed to the lactate demand in the community (see Figure 5c). This behavior is the first finding to resemble the reverse Warburg effect as an adaptive response to gradients. The simulation showed how the three subpopulations may respond to changes in their microenvironment. This feature was maintained on day 19 (see Figure 5d); however, the exchange interactions diminished as the oxygen approached zero.

A further metabolite that showed a peculiar behavior in response to the gradients was AKG, which had already been observed to interact with spheroid subpopulations under normoxic conditions, originating from proliferating cells and moving toward invasive cells. However, when approaching hypoxia, the community adapted in such a way that the directionality of this metabolite was inverted, generating the hypothesis that invasive cells have a better capacity to adapt to gradients and provide AKG to proliferating cells to compensate for tumor survival (see Figure 5e).

In the same way as with lactate, by day 19, the spheroid organization establishes an exchange of lactate from invasive cells to proliferative cells under normoxic conditions, and oxygen deprivation creates decay in the interactions; see Figure 5f. This behavior suggests that metabolic cross-feeding between proliferative and invasive cells is established in the first days of growth; day 6 shows a cellular organization that responds better to oxygen deprivation stress than day 19 since more invasive cells do not have cellular allies to help their survival.

### 2.8. Knock-Out In Silico of Metabolic Enzymes

To identify the reactions that could be essential for community growth, a knock-out (K.O.) essentiality analysis of metabolic reactions was implemented, including exchange reactions with the medium and individual intracellular reactions (see Section 4). Using growth rate values to determine the impact of K.O., a list of reactions whose absence directly affected the entire community was isolated. This suggested a dependence on metabolites from the medium, especially amino acids, as the most important for survival (see Figure 4b). The relationship was maintained on both days of growth, with the only difference being that serine might be essential for the day 19 spheroid.

To determine whether the metabolic dependence of the amino acids on the spheroid is robust in space, we repeated the simulations by mimicking the inner zone of the spheroids through the gradient of oxygen described in the previous section. Hence, we separately conducted a KO analysis by considering three feasible scenarios of oxygen: normoxia, mid-level, and hypoxia. The essentiality analysis of each oxygen environment did not show significant changes between days 6 and 19. These results postulate that, in this time scale, the metabolic phenotype of the entire community mainly depends on the amino acids and galactose as a carbohydrate present in the medium located in the external environment (see Figure 5). Despite the diversity of the components in the extracellular medium, such as pyruvate and some vitamins, the presence or absence of these components did not show importance in sustaining growth. Furthermore, no intracellular reactions appeared as essential in any of the three subpopulations, suggesting that the environment is the most probable mechanism altering the metabolic tumor phenotype.

Our in silico analysis highlights the importance of external amino acids for tumor growth. Despite some metabolites being essential for growth at any time and oxygen concentration (such as D-galactose, L-glutamine, and L-histidine), others have an interment relevance to the community growth rate; see Figure 5g. For instance, this is the case for L-valine, L-serine, and L-choline, whose relevance to the growth rate depends on time and oxygen availability. Overall, these results pave the way toward an in silico platform that can be used to construct testable experimental hypotheses in the design of therapeutic strategies, mainly focused on controlling the growth rate of 3D tumor spheroids.

Finally, the approximations performed with computational simulations were generated from information obtained from sequencing data. However, proteomic and metabolomic data are needed to ensure better results and to calibrate the reconstructions. This provides the presence or absence of metabolites that could generate false positives. The methodology proposed to perform these in silico analyses was preliminarily validated by bibliographic information that could qualitatively coincide with what is already known about breast cancer metabolism.

This in silico study of the inner metabolic phenotype in cancer provides a framework for designing testable experimental hypotheses and inspires its potential applications in oncological studies. With the advent of single-cell technologies, these computational methods will be essential for integrating data and generating hypotheses. Considering that cancer is a heterogeneous and communal structure, we envision that these methods will be necessary to move toward the design of treatments at a personalized level. Overall, these efforts allow us to conclude that cancer is more than one phenotype; instead, the presence of subpopulations contributes an additional degree of complexity, the inclusion of which should be addressed to improve the wellness of patients.

## 3. Discussion

In vitro, 3D tumor spheroids provide a perfect system to elucidate how intratumoral heterogeneity shapes metabolic organization and sustains the malignant phenotype at a community level. As shown in this study, our in silico analysis, which integrates single-cell RNASeq on MCF7, is a crucial strategy for achieving this aim. It allowed us to map the complex cross-feeding between the cellular components of the MCF-7 spheroid and predict how metabolic exchange changes their profile as a function of the relative cell abundances and time. This predictive power instills confidence in the model’s ability to anticipate metabolic changes, underscoring the importance of our in silico analysis in understanding the metabolic organization of 3D tumor spheroids.

Among the most remarkable metabolic exchanges, we highlighted the secretion and consumption of lactate. This metabolite is used as an energy source by cells in oxygen-restricted conditions, such as invasive cells inside the spheroid, and, from a diagnostic point of view, it has been listed as a biomarker of poor prognosis in patients with some solid cancers. This oncometabolite has been positioned as an essential marker due to its involvement as a tumor promoter and facilitator of immune evasion [25]. Our results broaden the picture of lactate involvement beyond tumor microenvironment regulation and as a potential energy source inside the inner subpopulations. Furthermore, in recent years, the reverse Warburg effect has gained increased relevance due to postulation surrounding the importance of lactate as an additional carbon source to produce biomass and not as a product, as described in the classical Warburg effect when cancer cells are near other neighbor cells. For instance, this is the case of the association between cancer-associated fibroblasts (CAFs) and tumor cells, which, as a system, exchange lactate and carbon sources to create a microenvironment that ensures cancer survival and progression [6]. Although the association has been experimentally induced between the tumor and its stroma, we supply evidence that a similar scenario could emerge among cells of the same tumor.

In agreement with some in vitro experiments, the promotion of lactic acidosis in the microenvironment by tumor cells can induce metabolic turnover in adjacent cells by promoting a shift from aerobic glycolysis to an oxidative phenotype [26]. The conditions simulated for invasive cells on day six could be equivalent to an experiment in which low oxygen levels and acidic conditions favor lactate consumption in breast cancer cell lines [27]. This behavior has also been observed in other tumor lines, such as lung cancer [27,28]. These results allow us to question whether this behavior is a hallmark of glycolytic tumors or a cellular adaptation in response to stimuli in their microenvironment. The organization proposed by the simulation for the spheroid on day 6 is valid when considering that proliferative cells condition the acidic microenvironment and that low oxygen levels promote the transformation of cells in the center of the spheroid to an invasive phenotype [29]. The metabolic connection among the subpopulations creates a principle of intratumoral cooperation to ensure the entire community’s survival.

In terms of the simulations of gradual oxygen deprivation in the microenvironment, we observed that the response of the invasive cells was to increase the rate of lactate consumption, and this was compensated for by the proliferative cells to the point that the reservoir subpopulation began to participate by secreting lactate to the medium; see Figure 5c. Our analysis supports the idea that this exchange is enhanced as a response to hypoxic conditions. However, the proposal on day 19 suggests that this association is already established and that oxygen gradients decrease the consumption rate of metabolites such as lactate; see Figure 5d. This may be related to the relative abundance of each subpopulation on day 19 because, on this day, there were more invasive cells than proliferative cells, which decreased the availability of lactate in the medium; the incorporation of pyruvate and the high rates of glutamine consumption compensated for this demand. Altogether, these findings suggest that the computational model on day six would have greater validity for future analyses evaluating the community’s response to oxygen deprivation. However, the day 19 model contemplates advanced tumor scenarios, where approximately 50% of the community is invasive. This could indicate the conditions needed in the culture medium to ensure the community’s survival. At this stage, the metabolism of the 3D spheroid is mainly based on the consumption of L-glutamine (Figure 5b), galactose (Appendix A), serine (Appendix A), and glycine (Appendix A). We are aware that our assumption that the three cell populations are well separated and arranged in layers is a simplification. The most probable scenario is that each layer combines the three subpopulations. In this context, two immediate goals should be addressed in future works: the presence of gradients as continuous variables and the possibility of a heterogeneous distribution of cells at radial coordinates.

Despite some simplifications in this work, we can postulate alternative metabolic pathways with a central role in tumor growth, among them being the activity of AKG exchange from proliferative cells to invasive cells on day 6. The role of AKG in cancer metabolism has been somewhat controversial. On the one hand, it has been observed to be an enhancer in radiotherapies by sensitizing triple-negative breast cancer tumor cells [30] and antagonizing tumor progression in P53-deficient malignancies [31]. In addition, AKG contributes to ammonium carriers through glutamate synthesis via reversible glutamate dehydrogenase, reducing urea production [32], and, in tumors such as colon cancer, AKG is associated with carcinogenesis [33]. Although glutamine is a semi-essential amino acid for tumor cells, the AKG skeleton is more soluble than glutamine, and its metabolic participation is usually similar to that of glutamine, compensating for the metabolic demands of invasive and proliferative cells. AKG can be directly incorporated into the TCA as succinyl-CoA through the action of alpha-ketoglutarate dehydrogenase to obtain energy via OXPHO [34], which would correspond to its participation in the metabolic synergy between proliferative and invasive cells as an alternative source of energy or as metabolic compensation since having a metabolite available in its most accessible form would be energetically more efficient. The ability of AKG to diffuse freely through voltage-dependent transporters facilitates its permeability, and its ability to diffuse freely through oxoglutarate/malate transporters without difficulty ensures its bioavailability in mitochondria for incorporation into the TCA [35]. Interestingly, a survey of the transport fluxes of AKG showed that its origin spans from proliferative to invasive cells. However, the effect of hypoxia created a switch in this behavior, reversing the roles. It is proposed that, under normoxic conditions, the proliferative cells provide AKG and lactate as metabolic sustenance to the invasive cells; this establishes a picture of metabolic compensation since this stage approximates the standard growth conditions of a spheroid. Figure 6 shows the proposed interaction between the two cell subpopulations with the most significant contribution to the sustenance of the community, which points to the presence of the reverse Warburg effect in cells of the same tumor type and a connection with the AKG that compensates for the energy rate of the cells in the center of the spheroid. The reversal of the dynamics, however, is associated with an increase in the rate of lactate production. Since it was maintained as a response to hypoxia and observed as a feature on day 19, this condition could be related to a marker of poor prognosis. Further study is crucial to elucidate the pathways involved in such associations and to explore how they can be intervened by selecting metabolic targets. The simulation suggested a connection between the reservoir and proliferative cells by exchanging fumarate and L-proline and providing glycine and adenosine triphosphate (ATP) in the medium available for the invading cells; see Figure 6. Despite the metabolic importance of these metabolites, the third subpopulation plays a crucial role in regulating tumor survival. The constant secretion of ATP, specifically in breast cancer, has been described to be involved in invasion processes to other tissues where ATP in the microenvironment could induce hypoxia-inducible factor (HIF1a), triggering epithelial–mesenchymal transition processes, even under normoxic conditions [35]. This makes this subpopulation a possible key player in promoting invasive processes. However, in tumor processes, the deficiency of the activity of enzymes such as fumarate hydratase (FH), responsible for the conversion of fumarate to malate, results in the intracellular accumulation of fumarate, and, in cases such as renal cancer, the accumulation of this metabolite inhibits the activity of prolyl hydroxylases, which leads to the accumulation of HIF1a [36]. The importance of the exchange of this metabolite lies beyond the regulation of HIF1a, the favoring of glycolytic activity over oxidative phosphorylation, and the regulation of the sensitization of reactive oxygen species [37] by orchestrating the antioxidant response via NERF2 when fumarate inhibits its regulator [38].

Given that lactate and AKG were not originally present in the growth medium, we performed additional tests to determine how important they might be in tumor growth. As shown in Appendix A, we can visualize additional tests the community underwent where lactate was included in the growth medium. This decreased doubling times compared to a scenario where lactate transport capacity was blocked. However, AKG included in the initial medium proved to be even better than lactate, bringing doubling times to about 14 h (Appendix A). In both scenarios, oxygen availability was simulated to decrease. Both metabolites were shown to maintain low doubling times, even with oxygen restriction. Taking these types of conclusions to experimental verification would be substantial to corroborate that the robustness tests used in the simulation are comparable to what occurs in an MCF-7 community and that the results provided by the simulation are approximations with some degree of validity.

In summary, the main objective of this project was to develop a strategy that could model the metabolic crosstalk among MCF-7 cell communities. The modeling implementation tests also allowed us to deduce and create a proposal for the interaction between the subpopulations of a spheroid. The proposal designed for the association of the three subpopulations corresponds to the growth period of day 6. In summary, the results suggest the existence of metabolic cooperation between proliferators and invaders and the possibility that the third subpopulation, the reservoir subpopulation, fulfills the function of providing energy in the form of ATP or fumarate. However, the reservoir subpopulation could prepare the conditions necessary to promote the transformation of invasive cells to metastatic processes, and the overstimulation of proliferative cells with fumarate could be an initial marker of transformation toward invasive phenotypes.

## 4. Materials and Methods

### 4.1. Data

The data for this study were retrieved from the Gene Expression Omnibus (GEO) under accession number GSE145633 [9]. The dataset consisted of two temporal points, day 6 and day 19, for MCF7 spheroids cultivated in L-15 medium (ATCC 30-2008). The researchers associated the three identified clusters with the following sample labels: Cluster A was linked to the invasive phenotype, Cluster B to the reservoir, and Cluster C to the proliferative phenotype; the cell labeling was conserved, and we took the author’s classification.

### 4.2. Data Imputation—sc-PHENIX

To recover some of the gene expression hidden due to the noisy nature of single-cell technology, we applied an imputation specialized for scRNA-seq data. We imputed the data using sc-PHENIX (single-cell phenotype recovery via non-linear expression imputation), a recent pipeline with classification performance capabilities currently reported in the literature [20]. The input matrix consisted of 363 cells and 23,923 genes. The sc-PHENIX method uses the count matrix and a UMAP space (https://github.com/resendislab/sc-PHENIX (accessed on 8 August 2024)). Imputation was performed to recover the missing gene expression from scRNA-seq and to improve differential gene expression. We used the same UMAP space work from the original work of the dataset [9]. The Sc-PHENIX parameters were t = 5, decay = 5, metric = ‘euclidean’, and knn = 20.

### 4.3. Reconstruction of Tissue-Specific Metabolic Network CORDA

Based on the imputed data for each subpopulation, we constructed a genome-scale metabolic model (GEM) using the Cost Optimization Reaction Dependency Assessment algorithm (CORDA) [13], identified on days 6 and 19 of progression. The CORDA is a method for reconstructing metabolic networks from a reference model (a database of all known reactions) and a confidence mapping for reactions. It allows for the reconstruction of metabolic models for tissues, patients, or experimental conditions from transcriptional or proteome data. The CORDA ensures the inclusion of as many high-confidence reactions as possible while minimizing the inclusion of absent reactions and maintaining metabolic requirements. It optimizes the identification of active metabolic reactions by balancing accuracy and cost. Using the single-cell RNAseq data from an MCF7 culture, we set up the metabolic activation for every cluster that triggers diverse cellular processes. To start the reconstruction, as input, the CORDA requires a confidence score for each reaction of the base model. We implemented an agglomerative cluster to obtain the confidence level for each gene in the data, fixing the number of clusters to 3. N-dimensional values were mapped considering the samples in each cluster, and the Euclidean distance was computed to associate genes with similar expressions. The confidence value ranged from 1 to 3, indicating low, medium, and high confidence scores. Additionally, we set the maximum confidence level for genes differentially overexpressed and the minimum confidence value for genes differentially underexpressed in each cluster; we considered differentially expressed genes as those with fold change ≥2 and *p*-value ≤ 0.05.

The base model for each genome-scale metabolic reconstruction was taken from Recon 2.2. It integrates 7785 reactions, 5324 metabolites, and 1675 genes [39]. In the absence of specific constraints for transport reaction fluxes in the medium, a generic margin commonly used in flux balance analyses was applied, allowing for minimal restrictions on the limits. However, for community and minimum media modeling, these limits were manually adjusted based on the experimental data setup [9] and tailored to the exchange ratios for the growth medium (Appendix A). To ensure an exchange of metabolites from the medium to the cytoplasm, we set the confidence level for the medium exchange reaction to 3. All codes were implemented in Python 3, which can be reproduced at https://github.com/resendislab/corda (accessed on 8 August 2024).

### 4.4. Analysis of Metabolic Reconstruction Quality

To ensure the reproducibility of the results in creating and utilizing genome-level metabolic reconstructions, we utilized MEMOTE (Metabolic Model Testing) [40]. MEMOTE serves as a tool for evaluating the quality of these reconstructions, offering a comprehensive report and an overall assessment. It scrutinizes aspects such as reaction stoichiometry, alignment with databases like BIGG or KEGG, and the incorporation of a biomass function. In this particular study, all reconstructions evaluated obtained a subtotal score of 94% due to suitable stoichiometry and reaction balance. However, the report showed a total score of 68% due to inconsistencies in the IDs with which the MEMOTE database makes comparisons. However, the simulation of communities with the implemented tool is fine for modeling purposes. The quality reports generated with MEMOTE can be consulted in Appendix A.

### 4.5. Community Modeling—MICOM

Community modeling was conducted by implementing the software MICOM (Microbial Community) version 0.34.1, a computational tool enabling the creation of cellular community models and the analysis of their metabolic phenotype [19]. Having obtained metabolic reconstructions for each subpopulation and defined the metabolites available in the media, MICOM consists of two sequential optimizations. One focuses on optimizing a community’s biomass, and the other is dedicated to identifying the best array so that all the subpopulations grow. In the first optimization, we define the community growth rate as the sum of all the subpopulation growth rates weighted by the relative abundance of each subpopulation. Importantly, this optimization is accomplished in steady-state conditions and constrained by thermodynamic, enzymatic, and environmental restrictions. More specifically, we resolve the fluxes that obey the following conditions:(1)$$\begin{array}{c} \mathrm{maximize}\;\mu_{\mathrm{community}}=\sum_{i}a_{i}\mu_{i}^{biomass} \end{array}$$(2)$$\begin{array}{c} \mathrm{subject}\;\mathrm{to}\;:\;S\cdot \nu =0 \end{array}$$(3)$$\begin{array}{c} \;\mu_{i}^{biomass}\geq \mu_{i}^{min} \end{array}$$(4)$$\begin{array}{c} \;lb_{i}<\nu_{i}<ub_{i} \end{array}$$(5)$$\begin{array}{c} lb_{ex}^{i}\leq a_{i}\nu_{i}^{ex}\leq ub_{i}^{ex} \end{array}$$(6)$$\begin{array}{c} lb_{m}^{i}\leq a_{i}\nu_{i}^{m}\leq ub_{i}^{m} \end{array}$$
where *S* and ν represent the stoichiometric matrix and the vector of fluxes in the metabolic reconstructions, respectively, and $\mu_{i}^{biomass}$ and $\mu_{i}^{min}$ denote the individual and minimal growth rates of subpopulation *i*. The relative abundance of subpopulation *i* is given by $a_{i}$. Equations (4)–(6) denote the thermodynamic constraints through the lower and upper bounds for all the reactions inside the reconstruction, the exchange with the extracellular media, and the exchange with the media, respectively. Overall, these conditions define the ‘community constraints’ and essentially calculate the steady-state behavior in terms of the external and media conditions. We applied a second optimization called the cooperative trade-off method to calculate the metabolic activity in the different subpopulations. This second optimization selects a sub-optimal phenotype of the maximum growth rate through a factor α, constrained to minimize the quadratic sum of the growth rates along all the subpopulations (L2 regularization). In other words,
(7)$$\begin{array}{c} \mathrm{minimize}\;\sum_{i}\mu_{i}^{2} \end{array}$$
(8)$$\begin{array}{c} \mathrm{subject}\;\mathrm{to}\;:\;\mu_{community}\geq \alpha \mu_{c}\;+\;\mathrm{community}\;\mathrm{constraints} \end{array}$$

We called α the trade-off parameter, which takes values from 0 to 1 and essentially mimics the degree of the competitive or cooperative phenotype observed along the entire community.

In this way, community models were generated from models in SBML (Systems Biology Markup Language) format with the capability to investigate exchange fluxes among different cellular populations. To perform community modeling on days 6 and 19, various factors were taken into account, including the growth medium specified in Appendix A, the three GEMs generated using the CORDA algorithm, and the relative abundance of each cellular subpopulation. On day 6, the relative abundances were as follows: invasive—0.0977, reservoir—0.3910, and proliferative—0.5113. On day 19, the relative abundances were as follows: invasive—0.5739, reservoir—0.2739, and proliferative—0.1522.

### 4.6. Essentiality Analysis by Knock-Out of Metabolic Reactions

This analysis integrates the community generated through MICOM and the functions provided by the COBRA toolbox [41]. In this process, a total count of the community’s reactions is performed, and a reaction-by-reaction knock-out (K.O.) evaluation is carried out. Optimization focuses on the objective function of the community, which is related to community growth. The community in question consists of a specific number of reactions, and a growth value is assigned to each reaction in response to the alteration of these reactions. The reactions that, when inhibited in silico, interrupt spheroid growth when the biomass value reaches zero or decreases below its optimal value for growth are considered essential.

### 4.7. Oxygen Gradients: Transport of Amino Acids and Other Carbon Sources

In the metabolic reconstructions, individual metabolic reactions had their fluxes restricted by defined upper and lower ranges, which corresponded to the directionality of the reactions. To simulate the community’s response to global changes in oxygen availability, the lower limit of the oxygen transport reaction was gradually adjusted, optimizing community growth in each scenario. This adjustment was carried out gradually, decreasing from 0.16 (normoxic conditions) in 0.01 unit decrements until reaching a value of zero, which corresponds to anoxia. The objective behind this modification was to evaluate the community’s behavior and determine the existence of metabolic cooperation to guarantee the whole community’s survival in the face of the stress induced by the lack of oxygen.

### 4.8. Flux Visualization

The output generated by the community model is presented in CSV format, and it contains a list of reactions along with the fluxes associated with each subpopulation. The total fluxes resulting from the simulation for day 6 and the total fluxes for the day 19 simulation (see Appendix A) are included. In the visualization of reactions related to metabolite exchange with the environment, a filter was applied that identifies reactions whose characteristic corresponds to ‘EX_metabolite_m’. This indicates that such reactions are related to population and growth medium exchange processes. Exchange reactions with the medium on day six and exchange reactions on day 19 are included (Appendix A). To visualize the metabolic fluxes of internal reactions, we manually filtered to select reactions involved in glycolytic metabolism, tricarboxylic acids, and galactose metabolism (Figure 4), and some involved in the pentose phosphate pathway and fatty acid oxidation (see Appendix A). According to the VMH database [24], metabolites were classified, and reactions were grouped according to their metabolic pathway. This visualization was performed by implementing the package ggalluvial from the ggplot2 library of R (version 4.2.2). All other figures were created with the Python matplotlib package (version 3.7.1); the information needed to make them can be found in detail in the GitHub (https://github.com/resendislab/Modeling_Heterogeneity_Cancer_Metabolism_MICOM (accessed on 8 August 2024).

## 5. Conclusions

In recent years, computational models have gained increased acceptance and popularity as a tool for exploring cancer metabolism and have tackled basic questions such as its adaptation to variations in the microenvironment. Our combined analysis with constraint-based modeling and scRNASeq data for MCF-7 spheroids allows us to observe possible pathways of metabolic interconnections between cell subpopulations. As experimentally reported, our model reproduces the presence of the reverse Warburg effect inside the tumor. Furthermore, it predicts additional or alternate pathways for energetic survival, being a great tool for suggesting more targeted designs for the study of cancer metabolism in cancer cell lines. Overall, our study contributes to enriching and improving the methods toward the rational search for the cornerstone that rules the metabolic alterations in cancer cell lines.

## Figures and Tables

**Figure 1 ijms-25-10894-f001:**
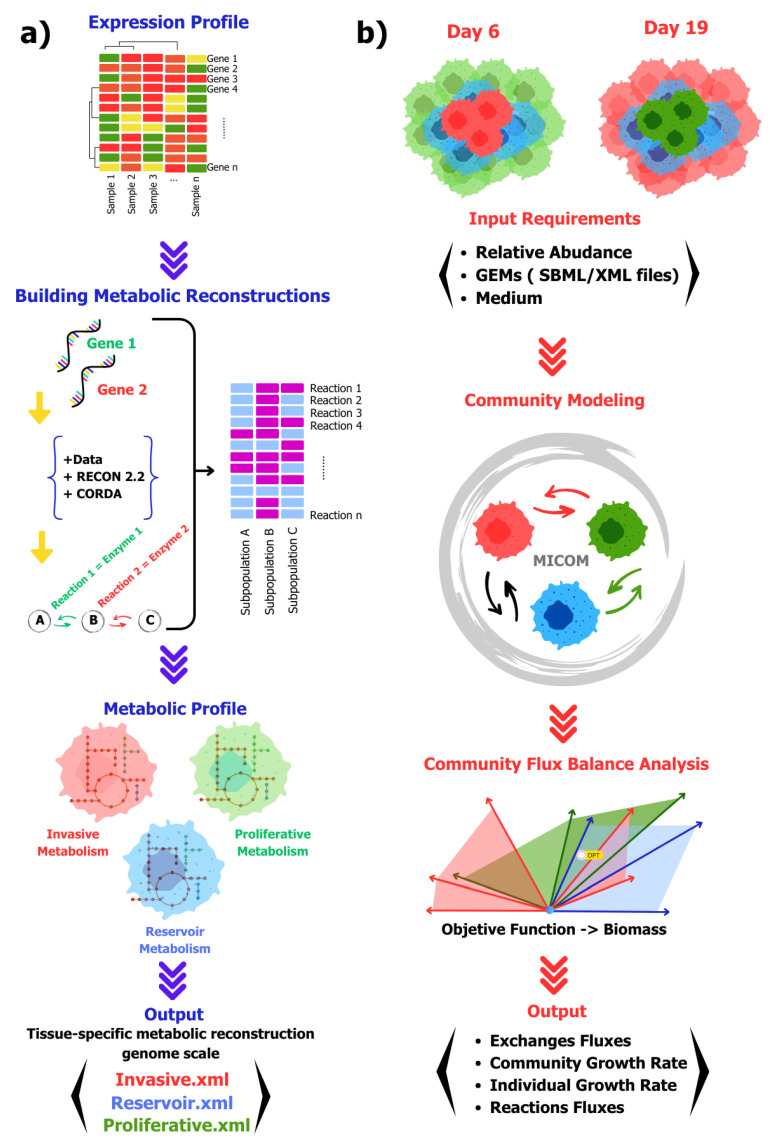
Workflow for community modeling from expression data. (**a**) Generation of genome-scale metabolic reconstructions. The analysis process started with obtaining scRNA-seq expression data from MCF-7 spheroids. Subsequently, the metabolic reconstructions were created by implementing the CORDA algorithm, assuming the presence of the metabolic enzymes based on their expression levels and using RECON 2.2 as a base template. The three cellular profiles previously described by Muciño et al. were generated, and, finally, three GEMs corresponding to each subpopulation were obtained. To ensure the quality of the reconstructions, they underwent an analysis using MEMOTE software version 0.17.0. (**b**) Community modeling and metabolic coupling. Part of the process involved gathering data on the community, including the abundance of each cellular subpopulation in the spheroid during the two days of analysis, the type of culture medium in which they grew, and a specific GEM for each phenotype. The community was constructed using the MICOM tool, which was used to optimize growth. As the final result, a report detailing the metabolic fluxes of each reaction in the individual reconstructions, as well as their interaction with the extracellular environment and population and individual growth, was obtained.

**Figure 2 ijms-25-10894-f002:**
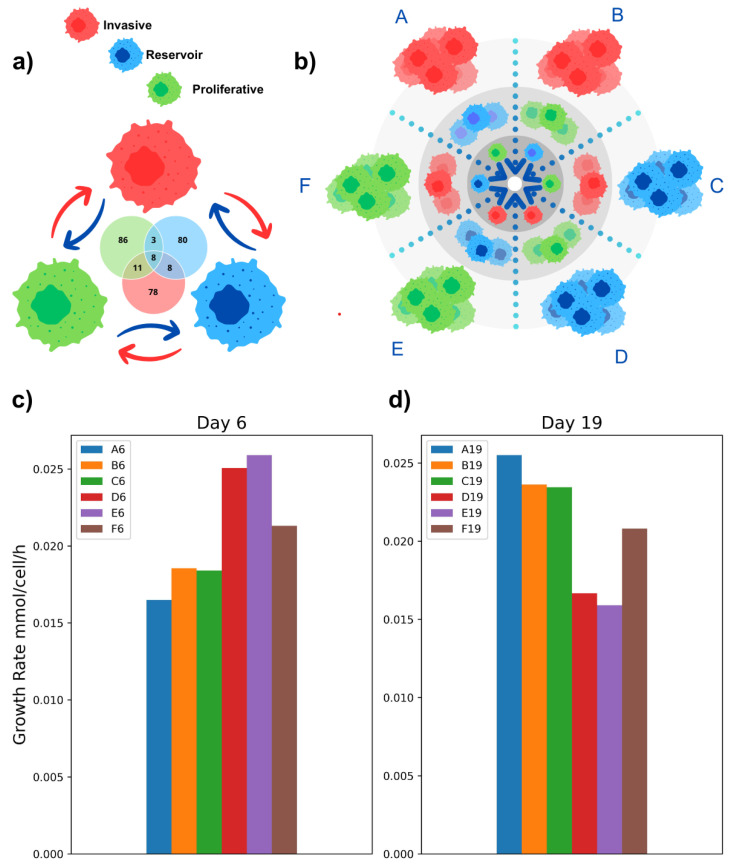
Spatial distribution simulation with oxygen gradients. (**a**) Graphical representation of metabolite exchange between each cell group in the community modeling. In the Venn diagram, the number of metabolites exchanged between and shared by the three groups collectively is shown in the central part, and the number of metabolites that they secreted to the extracellular medium is shown in the individual sections. (**b**) Proposed oxygen gradients for a three-dimensional model. The six combinations for the three subpopulations consider that each stratum change goes from its normoxia approach to levels close to hypoxia. (**c**,**d**) Community growth rates. The community rate measurement for each subpopulation shows that combination E on day 6 is the highest, and combination A on day 19 is the one suggested by the simulation. In both graphs, the Y-axis shows the community growth in flow units, and the X-axis shows the groups of combinations of possible cellular organization scenarios.

**Figure 3 ijms-25-10894-f003:**
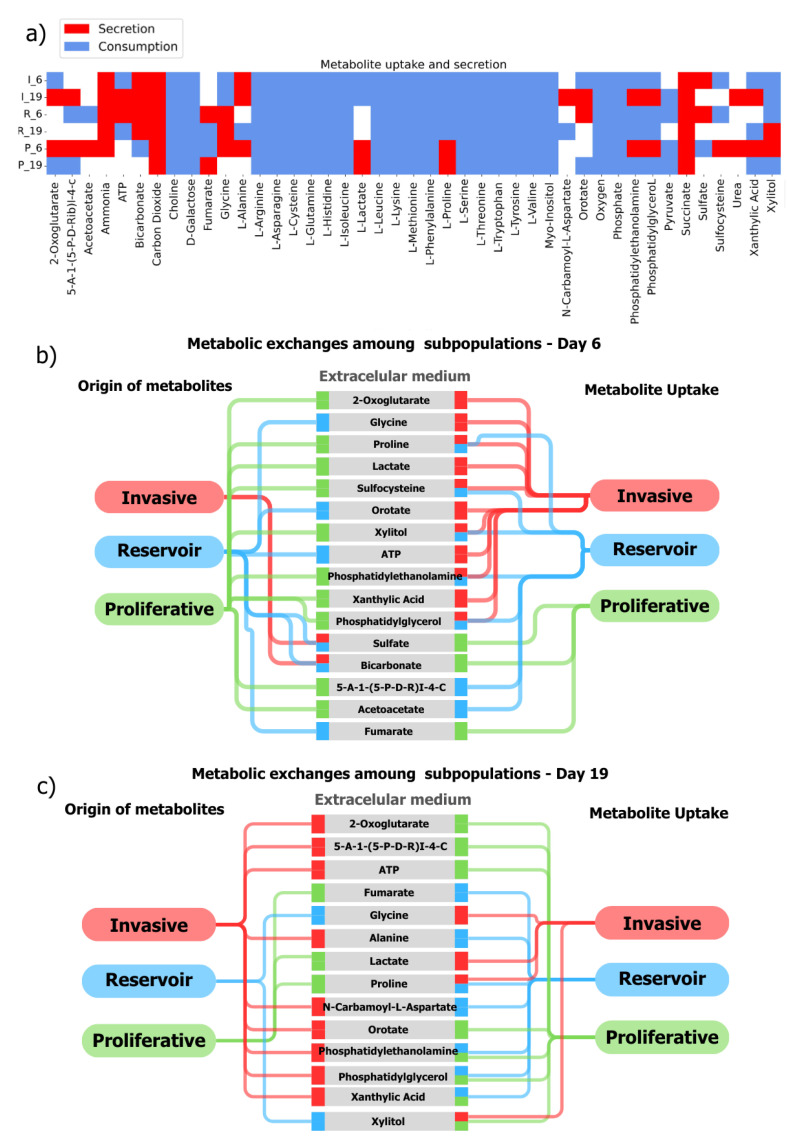
Metabolite exchange dynamics and Intracellular metabolism. (**a**) The secretion and consumption behaviors of the subpopulations. The consumption (blue) and secretion (red) behaviors of each subpopulation in the community were characterized by selecting the top 20 reactions with the highest fluxes on their respective days. Cells are identified by 6 or 19, along with their phenotype: I for invasive, P for proliferative, and R for reservoir. (**b**,**c**) Cross-feeding among subpopulations: The simulation suggests an exchange of metabolites absent from the initial growth medium. EX_ or exchange reactions with the highest fluxes in each subpopulation were filtered based on their direction (secretion or consumption). The left side of the panels shows the metabolite origin, the center shows its identifier (representing the extracellular medium), and the right side indicates its final destination—consumed by a remaining subpopulation. Colors represent the subpopulations: green for proliferative, blue for reservoir, and red for invasive. Although the same analysis was used, the two days of growth are shown with differing behaviors.

**Figure 4 ijms-25-10894-f004:**
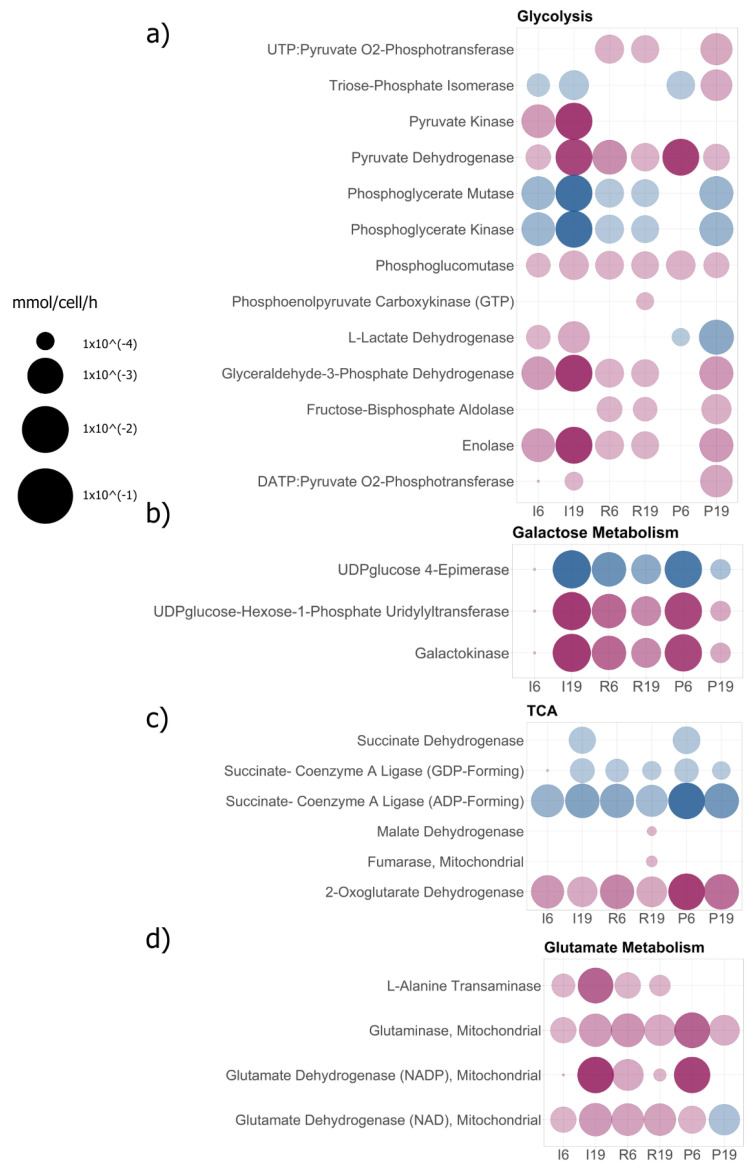
Enzymes with metabolic activity. The metabolic pathways were grouped according to the VMH database (see Section 4), in which each reaction is assigned a subsystem such as (**a**) glycolytic activity or glycolysis, (**b**) galactose metabolism, (**c**) tricarboxylic acid pathway or Krebs cycle, and (**d**) glutamate metabolism. Only reactions with activity in at least one of the three subpopulations were considered for visualization. To represent the directionality, a color identifier was used: the reddish color corresponds to the directionality of the reaction in its classical pathway described in the database, and blue corresponds to a reversal of the directionality of the reaction when observed. The blanks do not exclude the presence of enzymes in the reconstruction; they only reflect whether there was activity. The units of the FBA modeling correspond to the metabolic biotransformation rate, mmol/cell/hour, reflected in the size of the spheres for each reaction.

**Figure 5 ijms-25-10894-f005:**
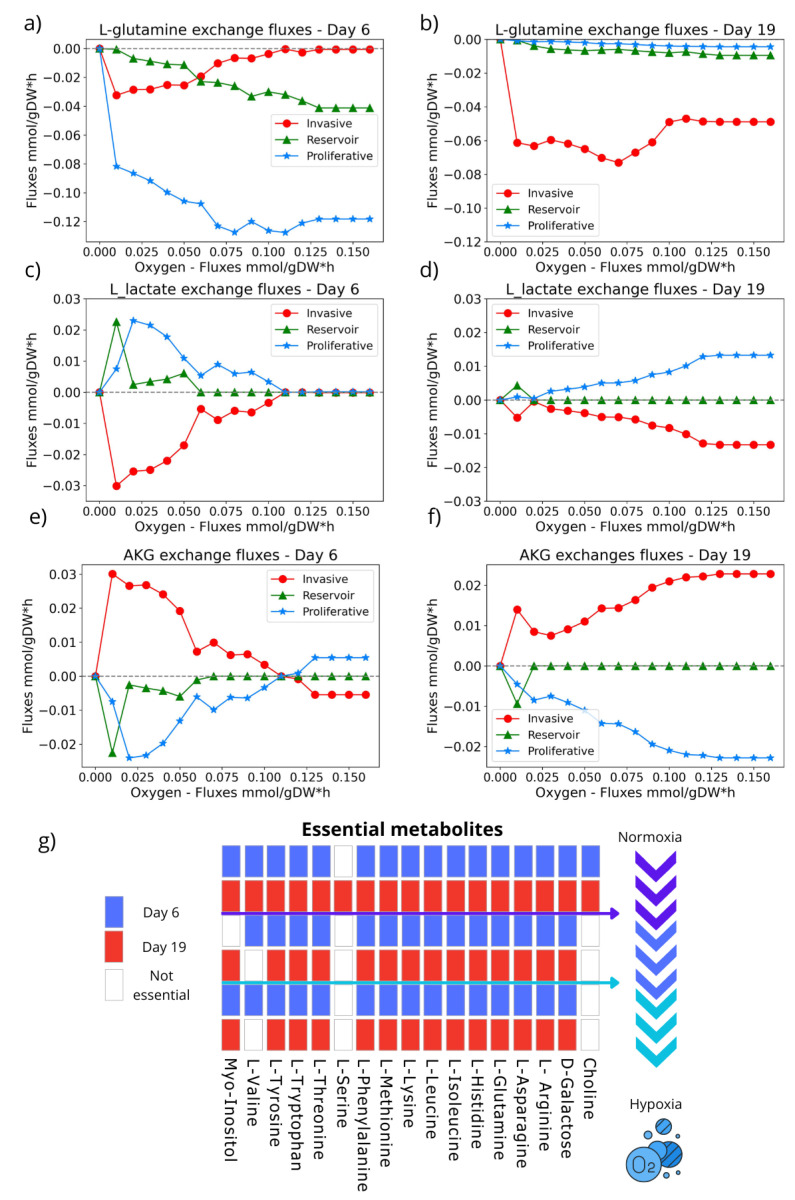
Community response to oxygen depletion and KO in silico. From right to left, the gradient from normoxia to hypoxia is represented, with oxygen fluxes decreasing. Each of the 16 marked points reflects the behavior of the subpopulations in terms of the consumption or production of a metabolite. Values above zero represent secretion, and those below zero represent consumption. Red lines correspond to invading cells, green lines to reservoirs, and blue lines to proliferative cells. Each point on the X-axis represents a different community simulation or scenario. As this is a deterministic scenario, we do not have any statistics since there were no variations in the repetitions of the simulations. To visualize the behavior of the community with the disposition of the elements of the medium, we selected metabolites that could be used as an energy source on different study days: (**a**) glutamine, (**c**) lactate, and (**e**) alpha-ketoglutarate on day six, and (**b**) glutamine, (**d**) lactate, and (**f**) alpha-ketoglutarate on day 19. (**g**) Essential metabolites in silico KO. The essentiality analysis generated a list of 17 metabolites whose absence in the medium proved lethal to the community’s growth. The state of the community, such as normoxia (upper part) or hypoxia (lower part), and a midpoint between the two, was recorded according to the growth day (red: day 6; blue: day 19).

**Figure 6 ijms-25-10894-f006:**
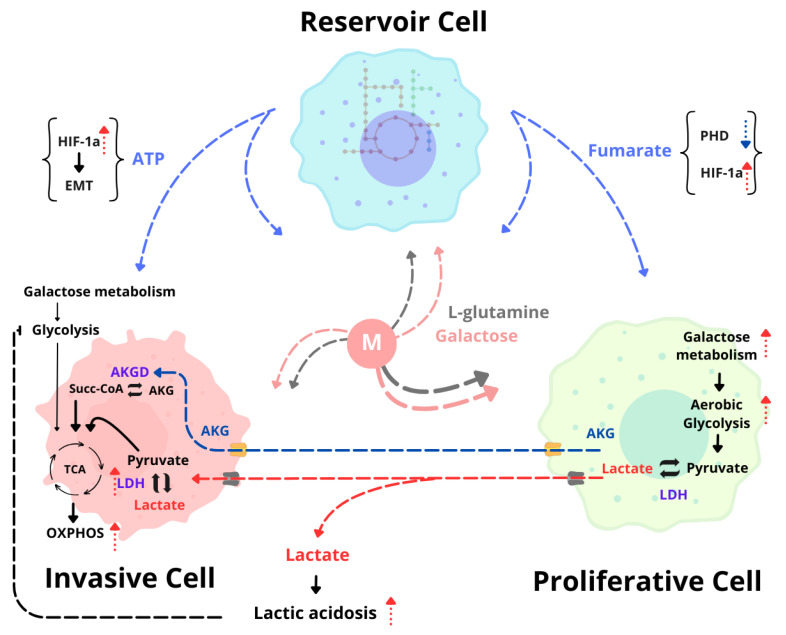
Metabolic interconnection between subpopulations (cross-feeding). A schematic representation of the interaction between the three cell subpopulations of the MCF-7 spheroid. Day 6 interaction is presented, where metabolism modeling suggested a connection between invasive (reddish) and proliferating (green) cells as metabolic sustenance. The primary exchange metabolites are lactate and AKG, which could be used as energy sources by the action of lactate dehydrogenase (LDH) and alpha-ketoglutarate dehydrogenase (AKGD). The lactate released into the extracellular medium (M) could induce lactic acidosis; this inhibits glycolytic activity (bibliographic information), which supports the cells’ use of alternative carbon sources. However, the third subpopulation (blue) or the reservoir shares high energy metabolites such as ATP, which, in addition to being energetic, could have secondary functions such as promoting the epithelial–mesenchymal transition (EMT) via the HIF1a pathway. This is similar to the effect of fumarate in regulating the inhibition of the HIF1a inhibitor prolyl hydroxylated (PHD). Both metabolites theoretically increase HIF1a expression (red dates) and promote transformation from a proliferating to an invasive spheroid, as observed via day 19 abundance.

## Data Availability

The data are available in the Gene Expression Omnibus (GEO) through accession number GSE145633, obtained from Muciño et al. The source code and simulation results supporting the findings of this study are publicly available at (https://github.com/resendislab/Modeling_Heterogeneity_Cancer_Metabolism_MICOM (accessed on 8 August 2024)).

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
