# Peer review of "Intratumoral Heterogeneity and Metabolic Cross-Feeding in a Three-Dimensional Breast Cancer Culture: An In Silico Perspective"

_ijms, 2024, doi:10.3390/ijms252010894_

Round 1

Reviewer 1 Report

Comments and Suggestions for Authors

The authors present a computational study of metabolic interactions in breast cancer cell spheroids. While the work is methodologically sound, I have serious concerns regarding the lack of novelty in the methodology and results. Additionally, the work presentation is misleading and lacks proper context and citation, as I will detail below.

Major Concerns:

  1. Claim of Novelty: The paper claims to have developed a novel "systems biology approach integrating single-cell RNASeq and genome-scale metabolic reconstruction for mapping metabolic cross-feeding." However, this claim is misleading, as it disregards prior work in the field, specifically the scFBA method developed in 2019 (Damiani et al., 2019, PLOS Computational Biology). The scFBA method was the first to integrate scRNAseq data into constraint-based metabolic models to study metabolic interactions, including the demonstration of the reverse Warburg effect. The authors’ failure to cite this foundational work is a significant oversight and misrepresents the originality of their approach.
  2. Lack of Explanation and Justification: The authors’ approach differs from scFBA, particularly in their use of the CORDA algorithm for tissue-specific model creation and MICOM for modeling cell-cell interactions. The CORDA approach largely differs from the pipe-capacity approach used in scFCBA and MICOM differently from scFBA seeks to optimize single-cell growth along with community growth. However. these differences are not adequately introduced or justified in the paper. CORDA and MICOM are mentioned without sufficient explanation, leaving readers without the necessary background to understand the rationale behind the choice of methods or how these choices impact the findings. Moreover, while the authors claim to extend these methods to cancer, it is unclear what novel contribution this work makes since they merely combine pre-existing methods rather than introducing innovations.
  3. Questionable Insights on the Reverse Warburg Effect: The authors suggest that their work provides new insights into the reverse Warburg effect, a phenomenon that has already been experimentally reported and computationally demonstrated using scFBA. The paper does not convincingly show that the observed metabolic cooperations are beneficial or that they arise from biological principles rather than being artifacts of the MICOM method. For instance, in scFBA, the removal of lactate exchange led to decreased collective growth, clearly supporting the reverse Warburg effect. The authors should perform similar robustness checks to ensure the biological validity of their findings.
  4. Misleading Title Regarding 3D Modeling: The title of the paper suggests the use of a 3D model, but the spatial aspect of the model is not evident. The work lacks a true geometric or spatial model and instead uses a rudimentary oxygen gradient, a concept already explored in popFBA (the precursor to scFBA, Bioinformatics, 2017). If the term "3D" refers to the experimental model rather than the computational approach, I suggest replacing "model" with "culture" in the title to avoid confusion.
  5. Unjustified Assumptions About Cell Population Arrangement: The assumption that the three cell populations are well-separated and arranged in layers, rather than mixed, is not sufficiently justified. The authors should clarify the biological relevance of this assumption and consider whether it accurately reflects in vivo tumor environments.
  6. Figure 3: Figure 3 is currently incomprehensible and requires a more detailed explanation.

Minor Concerns:

  • Figure 1 (Caption): The statement “Subsequently, using the CORDA algorithm, the presence of metabolic enzymes was inferred based on the expression data” is misleading. CORDA does not infer the presence of enzymes but assumes their presence based on high expression levels.
  • Line 345: The phrase “He designed” lacks clarity—please specify who is being referred to.
  • Line 606: The justification for restricting the range to [-1000, 1000] is unclear. Please explain which biological phenomena this range is meant to approximate.

Reviewer 2 Report

Comments and Suggestions for Authors

This study was a re-analysis of the RNA-seq dataset (GSE145633) previously  reported by authors using new methods. Recently, there have been reported that similarly analyze cell-cell interactions in the field using data obtained from spatial transcriptomics such as VISIUM. Without validating the author's analysis results, there is insufficient to judge the hypothesis was true or not.

I propose that authors will download the scRNA-seq dataset which had already analysed metabolic exchanges or major enzyme activity using other methods. Then it will be analyzed using author's new methods to obtain similar results. Again, please explain how to prove the analysis result is true.

Round 2

Reviewer 1 Report

Comments and Suggestions for Authors

The manuscript has been sufficiently improved to warrant publication in IJMS

Reviewer 2 Report

Comments and Suggestions for Authors

I fully understand the author's situation and explanations.